# Real-Life Effectivity of Dose Intensity Reduction of First-Line mFOLFIRI-Based Treatment of Metastatic Colorectal Cancers: Sometimes Less Is More

**Balázs Pécsi \* and László Mangel \***

Clinical Centre and Medical School, Institute of Oncotherapy, University of Pécs, H-7624 Pécs, Hungary
\* Correspondence: pecsi.balazs@t-online.hu (B.P.); mangel.laszlo@pte.hu (L.M.)

**Abstract:** Aim: The key purposes of the treatment of metastatic malignancies are to extend survival and maintain the quality of life. Recently it has been emphasized in the scientific literature that the maintenance of maximal dose intensity is not always beneficial. Method: We examined the effectiveness of first-line mFOLFIRI-based treatments used in mCRC indication in 515 patients, treated between 1 January 2013 and 31 December 2018 at the Department of Oncotherapy of the University of Pécs, on a basis of real-world retrospective data analysis. We studied the effect of decreased dose intensity treatment modifications on patient survival. Results: 45% of all patients achieved the optimal relative dose intensity (RDI) of 85%, and the median progression-free and overall survival (mPFS, mOS) were 199 and 578 days, compared to 322 and 743 days, (mPFS $p < 0.0002$, 1 y (year) PFS OR (odds ratio) 0.39 (95% CI: 0.26–0.56) and mOS $p = 0.0781$, 2 yrs OS OR 0.58 (95% CI: 0.39–0.85), respectively) in the group of patients not achieving the RDI of 85%. Conclusions: Decreased dose intensity did not reduce the effectiveness of treatment; in fact, there was a significant improvement in most of the analyzed parameters. The option of reduced dose intensity, which shows the same or even better results with less toxicity, should definitely be considered in the future palliative treatment of mCRC patients.

**Keywords:** metastatic colorectal carcinoma; first-line mFOLFIRI-based chemotherapy; dose intensity reduction; drug holiday; chemotherapy cycle-time

## 1. Introduction

One of the most serious oncological missions of the present times is the treatment of metastatic colorectal carcinoma (mCRC). The continuous increase in incidence (0.5–1.0% per year) is not the only concern, but also the continuous decrease in the average age of the patients affected (the number of people under 50 years increases by 2.2% every 5 years) and the increase in upfront metastatic cases (a 5% increase in 20 years) are also accentuated challenges. Despite the advantages of screening, modern diagnostics, and complex oncotherapy, mortality is still close to 40%. Approximately 20–50% of the patients detected and treated at an early stage become metastatic within 5 years, and as the average age of the population increases, the number of metachronous metastatic cases continues to increase [1,2].

Modern diagnostics and therapeutic methods have gone through a significant improvement in the care of mCRC, assisted by an expanding range of supportive methods. In addition to the conventional chemotherapy-based protocols, the use of biological and immune therapies is becoming increasingly focused. The development is clear, but in several areas, such as mCRC primary palliative treatment, "traditional" chemotherapy is still the backbone of cancer care. In the light of new data, the necessary revision and optimization of the conventional methods are essential.

As far as we know now, in potentially curative adjuvant treatments for early-stage diseases, obtaining the highest tumor-eliminating effect, and longer disease-free and overall

survival (DFS and OS) therapies on maximum dose intensity (DI) should be pursued. A decrease in DI in cases of treatment postponements, dose reductions, or the early discontinuation of treatment increases the chance of tumor recurrence and death. But in cases of metastatic or recurrent diseases, the importance of high DI is not unequivocally clear, and the published data are often contradictory. Classical cancer care principles recommend continuing palliative treatment up to progression or the limits of personal (psychical or somatic) tolerance. It is clear that in addition to the improved survival outcomes achieved by modern treatment methods, the toxicity of the active substances is not negligible. Full-dose treatment toxicity means a significant burden to the patient and may cause the unplanned interruption of effective treatment [3,4].

Recently, in the literature on mCRC treatment, a novel idea has emerged to query the real need for high DI chemotherapies in palliative cancer care. The most important messages are the following:

*" . . . findings suggested that reductions in RDI . . . due to treatment delays and dose reductions in response to adverse events do not necessarily lead to shortened TTF and OS"* [5].

*" . . . the need to reduce chemotherapy dosage due to side effects does not indicate a worse prognosis in our retrospective analysis. . . . this can in part be explained by better adaption to interindividual pharmacokinetics and a longer time of treatment"* [6].

*" . . . it seems unreasonable to try to maintain a high RDI, with a greater risk of toxicity, in palliative patients whose quality of life should be maintained as long as possible"* [7].

*"Maintenance (therapy) appeared to reduce cumulative grade 3/4 toxicity as compared to the continuous strategy while showing comparable efficacy"* [8].

*"The incorporation of treatment breaks and the use of staged treatment strategies appear to result in little or no detriment to overall survival. Treatment breaks also provide periods off chemotherapy that are highly valued by patients as well as resulting in a lower risk of significant toxicity"* [9].

*"The use of treatment de-escalation in mCRC is prevalent and these modifications do not appear to result in inferior outcomes."* [10].

The question is: which factors generally impede the maximal adherence to the original protocol scheme? The answer is simple, dose-limiting toxicity does. If we recognize that in a significant proportion of mCRC patients treatment cannot be carried out according to the protocol due to the observed side effects, then it is necessary to clarify whether these treatment modifications really influence the effectiveness of the treatment. Furthermore, if not, it is essential to keep on the maximum DI with all its risks. In the registration studies in the first-line mCRC setting, in the case of 30–76% of patients, it was necessary to alter the draft in the dose and/or treatment schedule. Together with these modifications, the effectiveness of the treatment scheme was demonstrated, however, most of these studies did not analyze the actual impact of DI on effectiveness. Some separated RDI data had diverse results on mPFS/OS [7,11–14].

To assess the real-life situation, we surveyed the effectiveness of mCRC palliative treatments over a six-year period. We made a retrospective real-world analysis of the consequences of protocol modifications with reduced DI of first-line mFOLFIRI-based treatment of mCRC patients. These findings, the related examples from the literature, and our results and conclusions are presented in this paper. Further results of the complex data collection of the survey will be published in another follow-up study.

## 2. Methods

We surveyed all ongoing first-line mFOLFIRI-based treatments at the Institute of Oncotherapy of the University of Pécs Clinical Centre between 1 January 2013 and 31 December 2018. We analyzed 25 different patient-related, disease-specific, and outcome parameters in every patient (age; performance status; treatment period and a number

of chemotherapy cycles; drug holiday(s) (treatment suspension over 56 days), time to progression and death; tumor localization and TN status; tumor grade; surgery of primary tumor; site and onset time of metastases; local treatment of metastases; the result of first restaging examination; the reason for discontinuing the treatment; degree and duration of dose reduction; exclusively). In the current work, based on these data, mToT (median time-on-treatment), mPFS and mOS values, the average cycle-time, and RDI values were determined and analyzed. Different dose-reduction patterns were analyzed, such as cycle-time extension and dose reduction. The effect of DI reduction on mToT/PFS/OS values was studied in 20% incremental cycle-time groups relative to the normal value (14 days). In the calculation of the average cycle-time, the duration of the drug holiday was not included in the ToT. The effect of RDI was evaluated in decreasing DI per 10% groups compared to normal (100%), and a direct comparison was made between RDI groups below and above 85% as recommended by earlier publications [4].

Descriptive statistics were used to characterize the patient cohorts. Differences in categorical parameters were analyzed using a two-sample *t*-test. Progression-free and overall survival were estimated using the Kaplan–Meier method. The level of significance of $p \leq 0.05$ was used. The odds ratio was calculated within 95% confidence intervals.

It should be noted that at the time of data analysis (17 March 2022), 40 patients (7.8%) were still alive.

## 3. Results

Of the 515 patients treated during the study period, there were total 8024 cycles of chemotherapy (median 12 cycles/patient, between 1 and 89 cycles). In total, 293 patients with mFOLFIRI (56.9%), 149 patients with bevacizumab-mFOLFIRI (28.9%) (BEV-FOLFIRI), and 73 patients with EGFR positive and RAS wild type mCRC with panitumumab/cetuximab-mFOLFIRI (14.2%) (PAN or CET, EGFRi-mFOLFIRI) were treated.

The permanent discontinuation of mFOLFIRI-based treatments without switching lines was due to progressive disease (52.4%), intolerable toxicity (hematological or intolerable side-effects; 20.4%), patients' rejection of further chemotherapy (12.4%), complete clinical remission or long-lasting stable disease (7.4%), and metastasectomy (systemic treatment after R0 metastasectomy was continued by "postmetastasectomy" FOLFOX scheme; 7.4%) (Table 1).

**Table 1.** Summary of chemotherapy-specific and survival data of the analyzed treatments, and comparison of these results with literary data.

| | | FOLFIRI | BEV-FOLFIRI | EGFRi-FOLFIRI |
|---|---|---|---|---|
| No. of Cases | | 293 | 149 | 73 |
| No. of chemotherapy cycles | | 4023 | 2796 | 1205 |
| Median No. of cycles per patient | | 12.0 (2–70) | 12.5 (1–81) | 12.0 (1–89) |
| Median cycle-time (days) | | 17.26 | 17.48 | 18.40 |
| Response rate (%) | | 16.0 | 41.0 | 52.2 |
| Median Time-on-Treatment (days) | | 179 (17–2010) | 237 (0–1834) | 213 (0–1654) |
| Median Progression-Free Survival (days) | | 241 (28–3274) | 310 (1–3297) | 267 (22–3556) |
| 1y PFS OR (95% CI) | | 0.50 (0.33–0.75) | | |
| | | | 1.35 (0.76–2.41) | |
| Median Overall Survival (days) | | 598 (46–3452) | 739 (54–3842) | 817 (71–3663) |
| 2 yrs OS OR (95% CI) | | 0.61 (0.40–0.95) | | |
| | | | 0.80 (0.44–1.47) | |
| Literary comparison | Median PFS (days) | 201–255 | 270–345 | 297–420 |
| | Median OS (days) | 529–654 | 700–772 | 605–1125 |
| | RR (%) | 50–56% | 43–63% | 47–67% |
| | Source | [11,12] | [15,16] | [17,18] |

The mPFS and mOS values of the treatments correspond to the values reported in the medical literature, only the mPFS value of EGFRi-mFOLFIRI treatments seems slightly lower, but this difference is no longer noticeable in the mOS value.

All patients were intended to be treated in accordance with professional protocols. All modifications were made due to the otherwise unmanageable toxicity or lack of personal tolerance. In most cases, these treatment modifications were temporary, but in some cases, the modifications turned permanent, due to necessity and professional decisions.

### 3.1. Impact of DI Reduction on Treatment Outcomes

Of the 515 treated patients, 195 (37.9%) had no meaningful DI changes (mean average cycle-time difference within +20% (<16.8 days), no dose reduction, no drug holiday). DI reducing effect (mean average cycle-time over +20% (>16.8 days) or dose reduction or drug holiday) appeared in 320 cases (62.1%). Compared to the unmodified treatment group, the occurrence of any of the DI reducing effects increases mPFS (180 vs. 321 days, $p < 0.0001$; 1 yr PFS OR 0.37 (95% CI: 0.24–0.55)) and mOS (564 vs. 758 days, $p = 0.0191$; 2 yrs OS OR 0.51 (95% CI: 0.35–0.77)) significantly. DI modifying effects were seen in combination in 174 cases (54.4%). When all three DI reductive effects were combined, mPFS and mOS increased to 783 and 1129 days compared to the unmodified treatment group ($p < 0.0001$; 1 yr PFS OR 0.02 (95% CI: 0.01–0.20) and $p = 0.0555$; 2 yrs OS OR 0.05 (95% CI: 0.01–0.41), respectively), however, the number of cases were only 12 (2.3%). Single modifications were observed as a cycle-time extension in 197 cases (38.3%), as a drug holiday in 24 cases (4.7%), and as dose reduction in 16 cases (3.1%). Compared to the unmodified treatment group, mPFS increased from 180 days in cycle-time extension group to 267 days ($p = 0.0195$; 1 yr PFS OR 0.59 (95% CI: 0.38–0.94)) and in drug holiday group to 382 days ($p = 0.0806$; 1 yr PFS OR 0.27 (95% CI: 0.12–0.66)). Median OS of these two groups increased from 564 days to 644 days ($p = 0.2762$; 2 yrs OS OR 0.69 (95% CI: 0.44–1.07)) and 851 days ($p = 0.1579$; 2 yrs OS OR 0.26 (95% CI: 0.09–0.73)), respectively. The changes in drug dose reduction group in mPFS (173 days, $p = 0.9927$; 1 yr PFS OR 0.72 (95% CI: 0.24–2.25)) and mOS (446 days, $p = 0.8511$; 2 yrs OS OR 0.44 (95% CI: 0.09–2.13)) were not significant. See Figure 1.

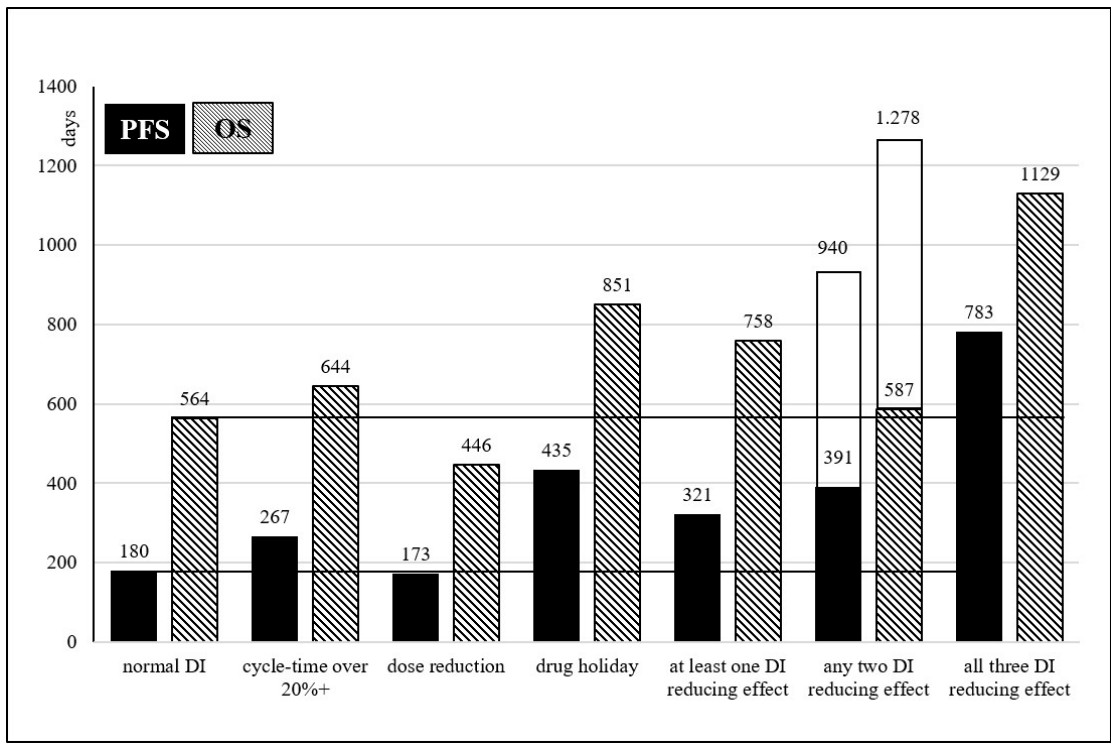

**Figure 1.** PFS/OS changes due to different DI reducing effects, separately and in combination, compared to the results of unmodified treatments.

*3.2. DI Reducing Deviations*

3.2.1. Cycle-Time Extension

During the observed period, a median of 12 and an average of 15.6 (1–89) cycles of chemotherapy per patient were performed. The cycle-time was maintained within a 10% deviation from the protocol (<15.4 days) in 21.6% of patients, and within 20% (<16.8%) in 46.2%. In all other cases (53.8%), we observed an average extension of the cycle-time above 20% (>16.8 days). The "cycle-time extension" was considered to be below 56 days, whereas above 56 days belonged to the "drug holiday" group. The causes of cycle-time extension (between 14 and 56 days) were toxicity, slow reconvalescence, diagnostics delay, scheduling conflict, or the personal request of the patient. In some cases, due to previously detected toxicity, we decided to extend the cycle-time permanently. The cycle-time extension was the most common DI-reducing modification of treatment (53.8%), compared to dose reduction (10.1%) and drug holiday (16.1%). Thus, cycle-time was the strongest factor in DI reduction in our series.

The data clearly shows that the increase in cycle-time with consequent RDI reduction—contrary to the expected decreased effectivity, resulted in a significant increase in effectiveness in the majority. Compared to cycle-time below a 20% extension, an increase of 40–60% cycle-time for mFOLFIRI and BEV-mFOLFIRI treatments showed a significant increase of mToT (163 vs. 209 days, $p = 0.0291$; 1 yr ToT OR 0.18 (95% CI: 0.06–0.48) and 141 vs. 421 days, $p = 0.0038$; 1 yr ToT OR 0.16 (95% CI: 0.06–0.47), respectively) and mPFS (184 vs. 348 days, $p = 0.0263$; 1 yr PFS OR 0.28 (95% CI: 0.12–0.63) and 238 vs. 540 days, $p = 0.0132$, 1 yr PFS OR 0.16 (95% CI: 0.06–0.47), respectively). However, only the BEV-mFOLFIRI comparison showed a significant increase in mOS (653 vs. 1127 days, $p = 0.0179$; 2 yrs OS OR 0.17 (95% CI: 0.04–0.65)). For EGFRi-mFOLFIRI treatments, the difference was not significant regarding any of the studied parameters (mToT 140 vs. 137 days, $p = 0.7259$; 1 yr ToT OR 0.41 (95% CI: 0.07–1.27), mPFS 137 vs. 224 days, $p = 0.7275$; 1 yr PFS OR 0.98 (95% CI: 0.20–4.79) and mOS 673 vs. 863 days, $p = 0.8826$; 2 yrs OS OR 0.92 (95% CI: 0.18–4.58), respectively). However, the combined evaluation of all three treatment groups showed significant increases in all values: mToT increased from 155 to 243 days ($p = 0.0003$; 1 yr ToT OR 0.19 (95% CI: 0.10–0.37)), mPFS increased from 202 to 378 days ($p = 0.0022$; 1 yr PFS OR 0.28 (95% CI: 0.16–0.50)) and mOS increased from 580 to 791 days ($p = 0.0526$; 2 yrs OS OR 0.49 (95% CI: 0.26–0.92)), respectively. See Figure 2A–C, Figures 3 and 4A.

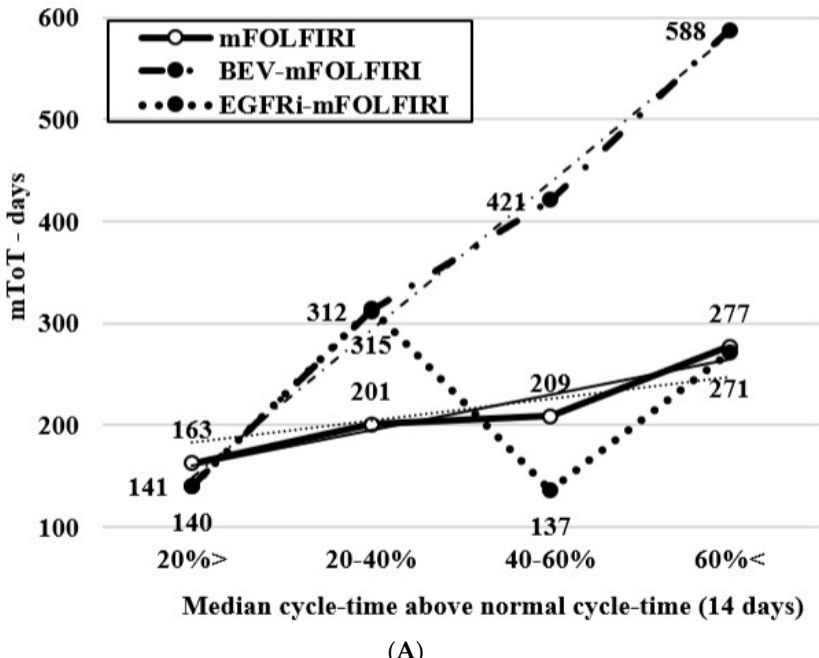

(**A**)

**Figure 2.** *Cont.*

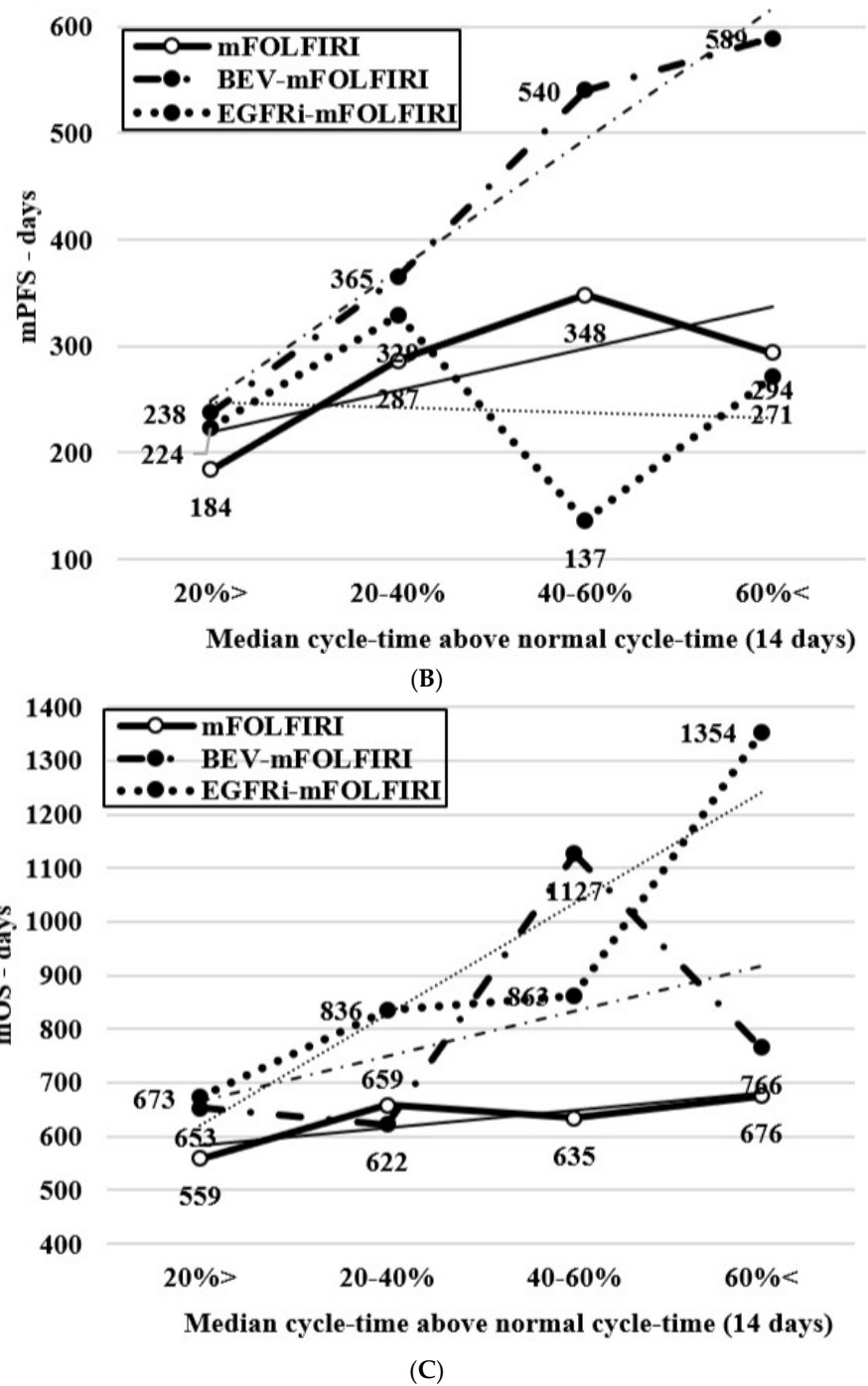

**Figure 2.** (**A**) mToT, (**B**) mPFS, and (**C**) mOS changes and trend lines according to the rate of deviation of cycle-time (incremental cycle-time groups of 20% relative to normal) by chemotherapy subgroups and trend lines.

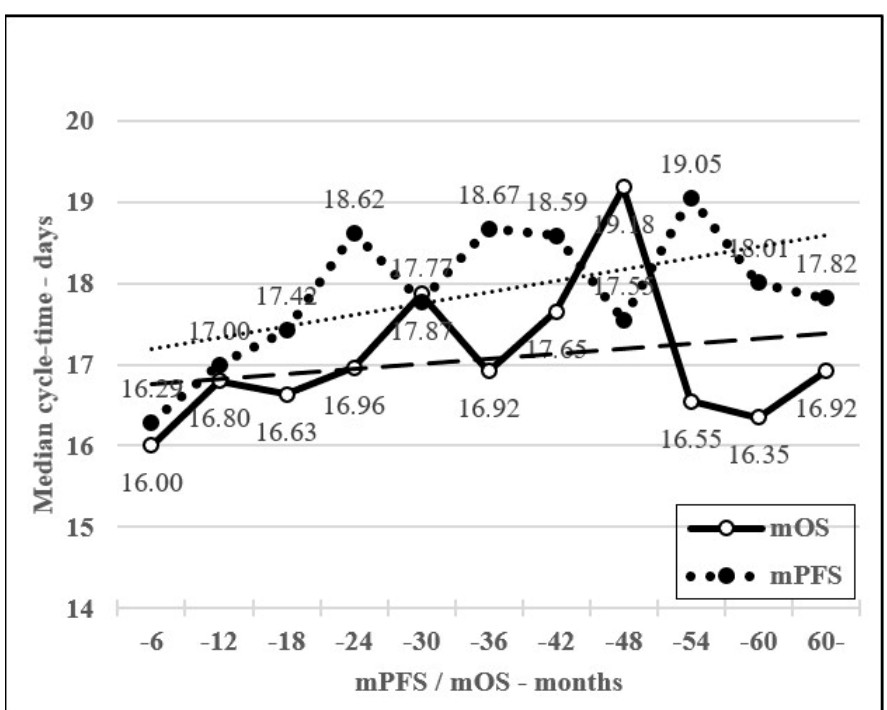

**Figure 3.** Median average cycle-time in incremental mPFS and mOS groups of 6-month subgroups and adherent trend lines.

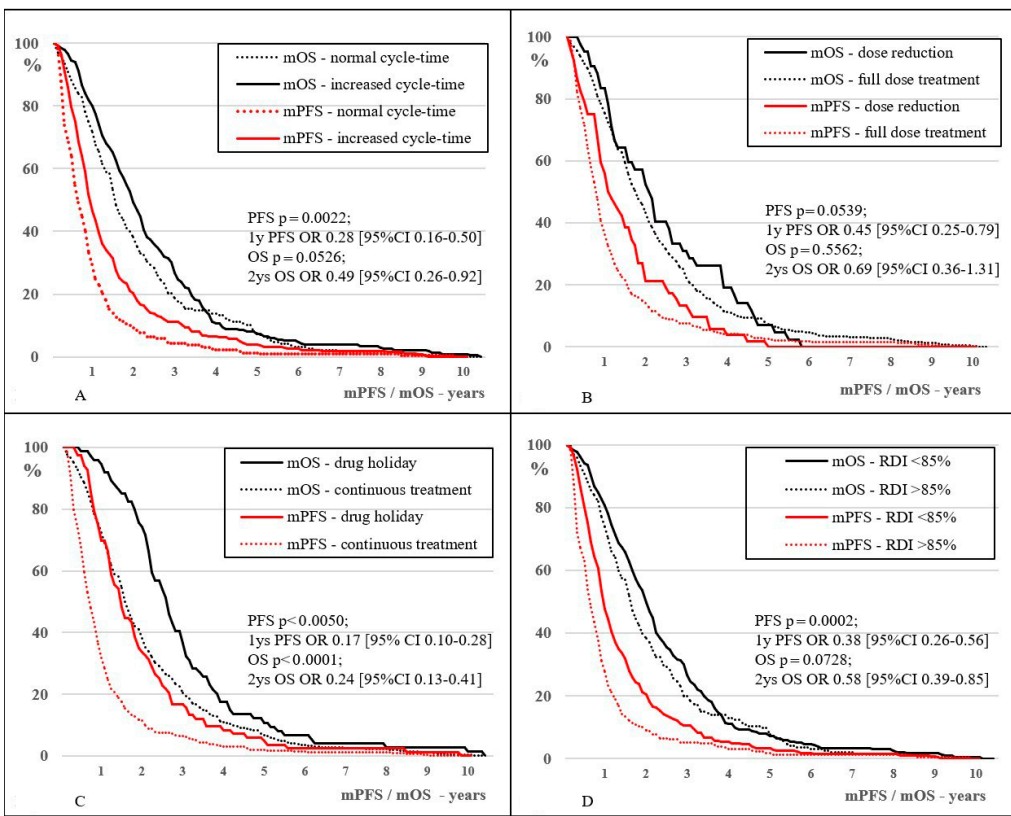

**Figure 4.** Kaplan–Meier curve for comparing mPFS and mOS of: (**A**) normal and extended cycle-time treatments (below or under 20% cycle-time extension); (**B**): full-dose and dose-reduced (any grade) treatments; (**C**): patients who had and a drug holiday and those who did not; (**D**): treatments with RDI above and below 85%.

### 3.2.2. Dose Reduction

Drug dose reduction or total drug termination was applied in 52 cases (10.1%). This was the least frequently used DI reductive method. Total drug shutdown of EGFRi occurred in 4 cases, after the median 17th cycle (4–28 cycles), and 5FU was stopped only in 1 case after the 4th cycle. The most toxic drug, IRI was stopped in 29 cases after the median 17th cycle (2–66 cycles). Dose reduction to prevent further toxicity occurred only for IRI, 18 patients had 10–30% dose reduction, 6 cases upfront and 12 cases after the median 10th cycle (3–53 cycles). The reason for upfront dose reduction was the presumed insufficient tolerance, and the later due to otherwise unmanageable toxicity. In cases of IRI dose modifications, the dose of 5FU and the dose of the targeted therapy drug remained unchanged. Compared to 259 and 657 days in the full-dose treatment group, in the dose reduced group mPFS and mOS were 393 and 773 days ($p = 0.0539$; 1 yr PFS OR 0.45 (95% CI: 0.25–0.79) and $p = 0.5562$; 2 yrs OS OR 0.69 (95% CI: 0.36–1.31), respectively). Dose reduction to maintain tolerance, therefore, did not reduce the effectiveness of treatment.

Furthermore, 18 patients (3.5%) had a dose reduction of 10–30% and 29 patients had complete IRI dose termination (5.6%). In these two groups, mPFS was 190 and 588 days ($p = 0.0096$; 1 yr PFS OR 0.95 (95% CI: 0.27–3.29)) and mOS was 583 and 819 days ($p = 0.0548$; 2 yrs OS OR 0.28 (95% CI: 0.07–1.20)), which compared to mPFS of 259 days in the control group without IRI reduction ($p = 0.6753$; 1 yr PFS OR 0.97 (95% CI: 0.36–2.63) and $p = 0.0026$; 1 yr PFS OR 0.92 (95% CI: 0.42–2.03), respectively). Only the complete termination of IRI treatment increased mPFS significantly.

The differences of mOS in different IRI dose reduced groups compared to the 657 days mOS observed in the control group were moderate ($p = 0.3263$; 2 yrs OS OR 0.60 (95% CI: 0.18–2.02) and $p = 0.1652$; 2 yrs OS OR 0.46 (95% CI: 0.20–1.09), respectively). The comparison between the IRI reduced-dose and IRI full-dose groups did not show a significant difference, probably partly due to the low number of cases. Based on this data, it is highly probable that the reduction or complete termination of the dose of the given high-toxic anti-cancer substance does not reduce the effectiveness of the treatment. See Figure 4B.

### 3.2.3. Drug Holiday

During the examined period, 83 patients (16.1%) had 98 events (1–3 occasions per patient) of drug holidays. These drug holidays had a median of 120 days and an average of 164 days (56–1027 days) treatment intervals. In all cases, the drug holiday was consensual with the patient, in cases of cCR or long-lasting (min. 4 months) SD. The reinduction of the same line chemotherapy was always due to observed progression at restaging examinations.

Compared to the mPFS of 227 and mOS of 595 days observed in the group of patients treated without drug holiday (but might have other DI-reducing effects), in the drug holiday group, mPFS and mOS were 540 and 975 days, ($p < 0.0050$; 1 yr PFS OR 0.17 (95% CI: 0.10–0.28) and $p < 0.0001$; 2 yrs OS OR 0.24 (95% CI: 0.13–0.41), respectively). In the case of drug holidays, not only primary mPFS, but also mOS increased significantly. Among the analyzed DI reducing effects, drug holiday had the largest effect on mOS ($p < 0.0001$). The effect of dose reduction and cycle-time extension did not, however the later almost reached the significant level ($p = 0.5562$ and $p = 0.0526$, respectively) on mOS. See Figure 4C.

### 3.2.4. Relative Dose Intensity (RDI)

The value of RDI was determined by the extension of cycle-time (53.8% of all patients had a cycle-time extension above 20%) and the rate of dose reduction (10.1% of all patients had some reduction). In the present patient population, the median RDI of IRI was 83.3% (2.9–100.0%) and the median RDI of 5FU, BEV, and EGFRi was 84.2% (46.6–100.0%). Considering each type of treatment, the median RDI of IRI and 5FU was 84.0 and 84.8% with mFOLFIRI treatment, 82.4 and 85.1% with BEV-mFOLFIRI treatment, and 81.6 and 81.7% with EGFRi-mFOLFIRI treatment [19,20].

The optimal RDI in medical literature recommendations was above 85%, and this value was achieved by 232 patients (45.0%). In this group, mToT, mPFS, and mOS were 151, 199, and 578 days, respectively, compared to 258, 322, and 748 days in the RDI below 85%, respectively; these differences were significant or nearly significant in all studied endpoints ($p < 0.0001$; 1 yr ToT OR 0.20 (95% CI: 0.12–0.34), $p = 0.0002$; 1 yr PFS OR 0.38 (95% CI: 0.26–0.56), and $p = 0.0728$; 2 yrs OS OR 0.58 (95% CI: 0.39–0.85), respectively). The 1 month increase in PFS was followed by an OS increase of 0.82 months in the RDI group above 85%, while in the RDI group below 85%, an OS increase of 0.96 months was recorded. Compared to a real-life survey in the medical literature, an OS increase of 0.68 months was recorded [4,21]. See Figures 4D and 5A,B.

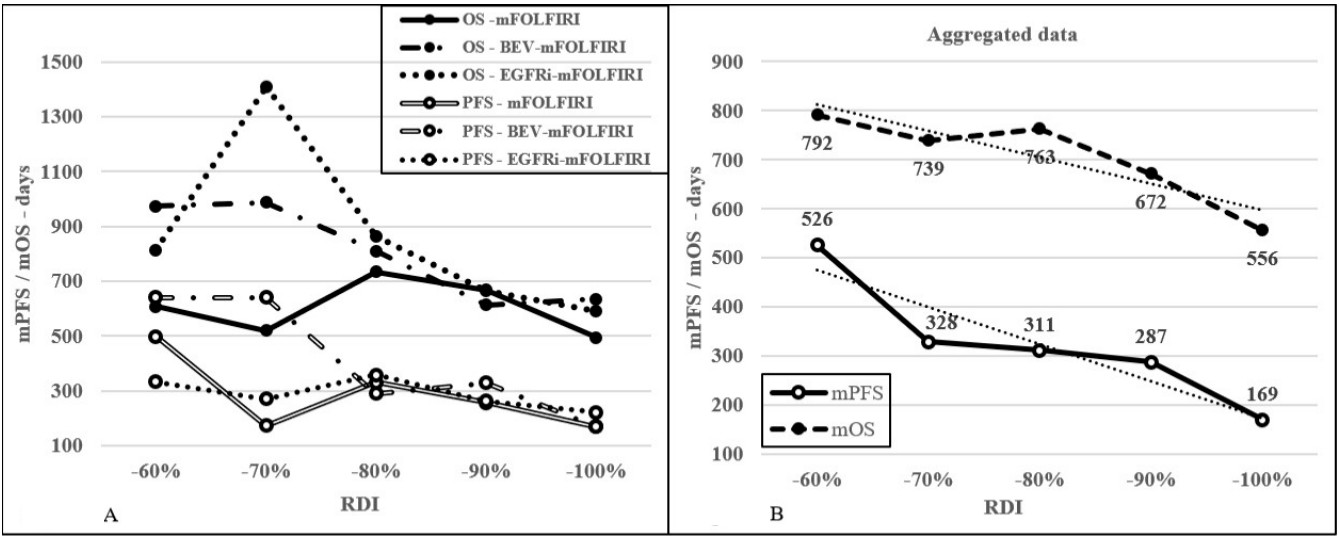

**Figure 5.** Effect of DI reduction (decreasing by 10% per group) on mPFS and mOS in each treatment type (**A**) and the aggregated data (**B**).

Based on the above data, the effects of DI reduction in the assigned patient population, analyzed alone and in combination with other factors, did not reduce the effectiveness of the treatments. It could be concluded, that there was a strong improvement in mTOT and mPFS, whereas a moderate improvement was observed in mOS.

## 4. Discussion

### 4.1. Elements of mFOLFIRI-Based Treatments

Since 1988, the most widely used administration of 5-fluorouracil (5FU) in CRC systemic therapy in Europe has been the 14-day cycle-time continuous infusion (CI) of the modified de Gramont scheme (mLV5FU). 5FU CI, which is effective throughout 1–21 days, showed significantly higher ORR and PFS compared to bolus dosage, with a moderate OS increase. The duration of infusional time appeared a more important factor than dose or DI [22,23].

Irinotecan (CPT-11, IRI) has several accepted dosing schemes (350 mg/m² q3w, 180 mg/m² q2w, and 100 mg/m² q1w; DI ≈ 100 mg/m²/week); different dosing and schedules did not demonstrate a significant difference in effectiveness ($p = 0.94$). Due to its unique pharmacokinetics, the 3-week dosage seems ideal, as toxic accumulation may occur in cases of shorter treatment periods [24–26].

The effect of bevacizumab (BEV) is not influenced by other anti-cancer drugs, as BEV does not influence their effect either. Nevertheless, synchronous dosing with oxaliplatin is necessary because of their confirmed synergistic effect. The elimination half-life of BEV is 18–20 days. The dosage regimens are fixed, no reduction accepted, 5–10 mg/body kgs every 2 weeks or 7.5–15 mg/body kgs every 3 weeks. Different dosage forms of BEV showed no significant difference in effectiveness [27].

Among EGFR inhibitors, the optimal dosage of cetuximab (CET) was observed with weekly dosing following the replenishment dose, the biweekly form was introduced later, when it was adapted to the de Gramont scheme. Pharmacokinetic studies showed that 21 days after administration of higher doses (above 100 mg/m$^2$), serum concentrations of CET were quantifiable. In the case of panitumumab (PAN), the biweekly dosage was registered, mainly due to the connection to the existing 2-week chemotherapy regimes. At the same time, appreciable clinical results can be achieved with PAN in the three-week application [28,29].

## 4.2. Obstacles to Cycle-Time Compliance

The scheduled, protocol-based continuation of long-term chemotherapy treatments is primarily influenced by dose-limiting toxicities. Grade 3/4 toxicities that could follow chemotherapy administration are the most common reasons for DI reductions (dose delays and dose reductions). Long-term or frequent Grade 1/2 side effects, causing physical intolerance and insufficient compliance, are also not negligible, they also lead to undesired dose reductions and delays, or even temporary or eventual interruption of the treatment.

Chemotherapy-induced neutropenia (CIN) and the consequent life-threatening febrile neutropenia (FN) are the most important dose-limiting factors. After chemotherapy, depending on the active substance and its dose, neutrophil nadir occurs after 10–14 days and it takes another 10–14 days to recover with new, mature neutrophil cells. The first CIN was detected after the first cycle in more than 50% of cases. After the first cycle, Grade 3/4 CIN occurred in 19.3%, which led to forced DI reduction, resulting in 74% RDI. The frequency of Grade 3/4 neutropenias in mCRC treatments is as follows: de Gramont scheme 4%, mFOLFIRI 27–31% and BEV-mFOLFIRI 26–37% [30,31].

The other significant dose-limiting side effect is diarrhea (chemotherapy-induced diarrhea, CID). Acute diarrhea is a dose-dependent cholinergic reaction, while late-type diarrhea occurs beyond 24 h (median 5 days), with a more severe course, and its dose-dependency is debated. The frequency of Grade 3/4 diarrhea in mCRC treatment is as follows: de Gramont scheme 8%, mFOLFIRI 21–44% and BEV-mFOLFIRI 32% [32,33].

The degree of side effects shows significant interindividual differences. Genetical changes in certain genes (e.g., DPYD, UGT1A1 gene-mutations), which encode enzymes that play a key role in the elimination of active substances, may involve up to 10% of the population, and can significantly decrease enzyme activity, which increases the risk of toxicity, even to fatal levels [23,34].

In conclusion, despite the proven early benefits of the de Gramont scheme, which is the basis for mCRC treatment in Europe, it is maybe not the only appropriate 5FU dosing scheme in today's practice. There is no clinical data available on the effect of 5FU CI with a longer cycle-time. The unchanged original scheme was supplemented by new-generation drugs (IRI, OXA, VEGFi, EGFRi) connected to the biweekly regime. The increase in the effectiveness of these combined schemes could reduce the importance of the basic 5FU CI. The DI of separate 5FU treatments was not significantly affected due to the 4–8% frequency of CIN and CID, unlike the 21–37% Grade 3/4 toxicity observed with IRI treatments. The most toxic component of mFOLFIRI-based mCRC treatments is IRI. In the case of IRI, in principle, 21-day dosing would be preferable, as this dosing can adapt to the natural neutrophil circle. For BEV and EGFRi, the two- and three-week dose is equivalent. Grade 3/4 toxicity may be detected with normal dosing in a quarter to a third of patients, which has a significant effect on further treatment. Long-term or frequent mild toxicity does not require treatment modification, but may reduce the patients' compliance, which may further lead to treatment modification.

## 4.3. Consequences of Decrease in DI

Due to significant interindividual differences (such as body fat ratio, enzyme activity, and bone marrow tolerance), the currently used schematic treatment principles and dose calculation methods are far from the desired personally tailored determination of optimal

treatment [20,35]. Unrecognized under-dosing may appear in up to 30% of cases, while due to differences in the elimination of cytotoxic agents (up to 4–50 times), it is common to achieve Grade 3/4 dose-limiting toxicity (21–37%) [36].

According to some reviews, keeping RDI above 85% is a criterion of optimal PFS and OS. An RCT with a high number of cases reported significant improvements of RR (65 vs. 6%, $p < 0.01$), PFS (9.9 vs. 5.6 months, $p < 0.01$) and OS (26.7 vs. 12.9 months, $p = 0.01$) with RDI above 85% [3,37]. Although another RCT confirmed that a high RDI value IRI has a significant PFS advantage ($p = 0.03$; HR 0.58 (95% CI: 0.36–0.95)), however, this advantage was no longer noticeable in OS ($p = 0.18$; HR 0.72 (95% CI: 0.45–1.17)). In both RCTs, a higher cycle count was observed in addition to low RDI, which meant nearly the same cumulative total dose administration [7]. Other opinions suggest that forced dose reductions and delays are necessary for palliative treatments due to high toxicity, but these have no significant negative effects on PFS and OS [38,39]. Switching active chemotherapy to maintenance, intermittent treatment, observation without treatment does not worsen OS, unlike the continuation of active full-dose therapy [8,40].

Although the conclusions of each RCT above are about the effectiveness of the registered protocol, dose modification in the treatment of mCRC affects a significant proportion of the cycles given. In some RCTs, DI reduction also affects 34–85.4% of patients and 13–30.3% of cycles administered, of which these modifications primarily mean dose delays (19.3–25.9% or up to 86%), and secondly dose reductions (8.2–24.4% or up to 66%). As a result, the median RDIs were 73% and 86.7%, respectively [8].

### 4.4. Importance of Maintenance Treatments

There is no clear recommendation as to which phase of palliative treatment may provide the possibility of treatment modification. Some authors suggest considering it while observing persistent or recurrent high-grade toxicity, other authors recommend rethinking the further thematics of treatment in the case of cCR or long-lasting SD results without regression.

The dosing options are the following:

* Continuous therapy—unchanged, full-dose chemotherapy;
* Maintenance therapy—dose reduction or discarding the highly toxic drug component;
* Intermittent therapy—chemotherapy-free intervals, CFIs;
* Drug holiday—stop the therapy till progression [41].

Several RCTs and meta-analyses involving significant numbers of patients emphasizes the need to compare the options. Maintenance therapy (fluoropyrimidine, FP) (FP, FP + BEV or BEV) is preferable regarding PFS compared to CFI (HR 0.53 (95% CI: 0.40–0.69)), to observation (HR 0.53 (95% CI: 0.40–0.69)) and to the continuous treatment ($p < 0.0001$; HR 0.62 [95% CI 0.51–0.75)). Another study found that maintenance therapy is equivalent to continuous treatment (HR 1.18 (95% CI: 0.96–1.46)). CFI is more advantageous (HR 0.53 (95% CI: 0.44–0.64); $p < 0.0001$), while observation is equivalent (HR 0.71 (95% CI: 0.46–1.09)) to continuous treatment in terms of PFS. A meta-analysis declared that compared with observation, maintenance therapy shows a PFS benefit (HR 0.58 (95% CI: 0.43–0.77)), but not an OS benefit (HR 0.91 (95% CI: 0.83–1.01)). From the point of view of toxicity, including tolerance, each form of modification is more beneficial than continuous treatment. All types of maintenance therapies showed a significant PFS advantage over observation. Maintenance therapy (FP and FP + BEV) showed the highest probability of improvement in PFS (67.1% FP, 99.8% FP + BEV) and OS (81.3% FP, 73.2% FP + BEV). The continuation of upfront full dose cytotoxic therapy until progression without maintenance or observation periods was not proven to be beneficial. Maintenance therapy (FP ± BEV) with considerably lower toxicity, significantly improves PFS without affecting OS [8,10,40].

The above studies have clearly demonstrated that lower RDI treatments, resulting in less toxicity, and treatment-free periods do not cause undesired reductions in PFS or OS. It is of great value in itself if the same oncological effectiveness can be achieved at a lower burden of toxicity. In medical literature, we have not found any relevant sources regarding

the relationship between cycle-time and ToT/PFS/OS, and this is believed to be the first retrospective analysis on this topic. However, based on the published results on RDI, it can be assumed that a similar RDI with a similar rate of dose reduction was achievable by the same rate of cycle-time extension.

The development of mCRC treatments in Europe was based on 5FU CI in accordance with the de Gramont scheme. The de Gramont scheme was spread mainly due to its early results (ORR, PFS) as a form of optimal dosage of 5FU. Despite its evident results, several other schemes of 5FU (Roswell Park, Mayo) are still in everyday use, mainly in America and Asia. The novel active substances were "automatically" adapted to the de Gramont scheme after phase II. RCTs, despite the fact that their use in longer cycle-time is equally effective, are less toxic. This means that keeping the 14-day cycle-time is considered to be important to maintain the de Gramont scheme, which is the less toxic component of the mFOLFIRI-based treatments. In the case of IRI, the 3-weekly administration seems ideal, and for the other active substances (VEGFi and EGFRi), a longer cycle-time is also an effective option. The effect of 5FU CI was not examined with a longer cycle time. The high rate of toxicity of each combination in a significant proportion of patients makes it impossible to continue treatment fully according to protocol, requiring DI reductions (dose reduction, cycle-time extension). The 14-day cycle-time, mainly due to frequent hematotoxicity, seems "too short": it is shorter than the natural neutrophil cycle time after administration of chemotherapy, so the phenomenon of cumulative toxicity is becoming an increasingly important topic, as it is affecting a great number of patients.

Even though mainstream international medical literature still links the optimal effectiveness of palliative treatments to maintaining high RDI, according to several recent meta-analyses, this is only justified in cases of conversion treatments. The summary data, in contrast to continuous treatment, have already demonstrated at least the equivalence of maintenance treatments, but drug holidays may also be equally effective with indisputably less toxicity. Although the available clinical trials do not specifically analyze cycle-time changes, it can be concluded from the dose reduction rate and the RDI values that these results were only achievable with approximately the same delay in average cycle-time, similar to those observed in our study. The idea of effective low-dose treatments is supported by the theory of "competitive release" that challenges to the traditional "maximum dose density" paradigm of tumor treatment [42].

In addition to the forced reduced DI treatments, a higher number of treatments, i.e., higher ToT, was observed, while longer PFS was also confirmed. Longer PFS generally means longer OS as well. The longer ToT automatically brings with it a further cycle-time extension, as a result of the increasing frequency of treatment postponements due to the decrease in somatic and/or physical tolerance during a longer treatment period. A high number of patients were treated with significantly reduced DI without any disadvantages. This rightly raises the question of whether treatments with upfront DI reduction (at least longer cycle-times) can achieve the same effectiveness (PFS, OS) compared to unchanged protocols. If yes, then the same result with lower and less frequent toxicity, as toxicity is one of the major burdens for patients, is more advantageous and more tolerable in chronic treatments. According to the results of our study, this is a realistic possibility. We think that instead of toxicity forced additional DI reductions, with upfront longer cycle-times, based on further data analyses and RCTs, equivalent oncologic results may be achieved.

To confirm our findings, a further opportunity would be the reassessment of previous clinical trials and the data of real-life publications; moreover, a prospective, randomized clinical trial is required. The results may even lead to changes in the "gold standard" care models.

## 5. Conclusions

Our data suggest that unplanned lower DI treatments in first-line mFOLFIRI-based chemotherapy of mCRC patients have no undesired effect on PFS and OS. DI reduction was a result of toxicity, and appeared mainly in the form of cycle-time extension, less

frequently in dose reduction. These forced changes led to significant DI reduction causing underdosed treatment. At present, the optimal effectiveness of palliative treatments is thought to be connected to maintaining high RDI. However, according to our data and other authors' opinions, the trend is the opposite: the result of DI reduction significantly improves PFS, and makes moderate improvements in OS as well. DI reduction is parallel to toxicity reduction, which means less burden and a better quality of life for the patients.

**Author Contributions:** Conceptualization, B.P. and L.M.; methodology, B.P.; investigation, B.P.; resources, B.P.; writing—original draft preparation, B.P.; writing—review and editing, L.M.; visualization, B.P.; supervision, L.M.; project administration, L.M.; funding acquisition, L.M. All authors have read and agreed to the published version of the manuscript.

**Funding:** This work was partially supported by the European Union and the Government of Hungary under the Grant GINOP-2.2.1-15-2017-00067 (Networked Analytical Opportunities and Data Utilization in Health Care, Survey of the Effectiveness of Modern Radiotherapy Technologies and General Cancer Care). Program participant: László Mangel.

**Institutional Review Board Statement:** Data collection was permitted by the responsible Hungarian Authorities (ETT-TUKEB, IV/7266-1/2020/EKU).

**Informed Consent Statement:** Non-interventional, anonymized retrospective data collection was permitted by the responsible authorities.

**Data Availability Statement:** The data presented in this study are available on request from the corresponding author. The data are not publicly available due to personal data protection.

**Acknowledgments:** Authors would thank to Henrik Sárkány and Tibor Holcz for their IT ideas and for the statistical and graphical support.

**Conflicts of Interest:** The authors declare no conflict of interest.

### List of Abbreviations (in Alphabetical Order)

5FU—5-fluoro-uracil, BEV—bevacizumab, CET—cetuximab, CFI—chemotherapy-free interval, CI—continuous infusion, CID—chemotherapy-induced diarrhea, CIN—chemotherapy-induced neutropenia, DFS—disease-free survival, DI—dose intensity, EGFR—epidermal growth factor receptor, FN—febrile neutropenia, FP—fluoropyrimidin, IRI—irinotecan, m—median, mCRC—metastatic colorectal carcinoma, OR—odds ratio, OS—overall survival, OXA—oxaliplatin, PAN—panitumumab, PFS—progression-free survival, RAS—rat sarcoma oncogene, RCT—randomized controlled trial, RDI—relative dose intensity, ToT—Time on Treatment.

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
