# Peer review of "Real-Life Effectivity of Dose Intensity Reduction of First-Line mFOLFIRI-Based Treatment of Metastatic Colorectal Cancers: Sometimes Less Is More"

_curroncol, doi:10.3390/curroncol30010069_

Round 1
Reviewer 1 Report
This is a retrospective study which investigated the effect of decreased dose intensity treatment modifications on patient survival in patients with metastatic colorectal cancer treated with first-line mFOLFIRI-based treatment. The authors showed decreased dose intensity did not reduce the effectiveness of treatment, in fact, there was a significant improvement in most of the analyzed parameters.
This study was conducted well, and the methods are appropriate. The data are presented clearly. In general, this is a well-written paper that presents interesting data. The results will be of interest to clinicians in the field.
However, the following a minor issue require clarification:
Minor
1. (P8L267) “Figure 6” should be changed to “Figure 5”.
2. Limitations in this study should be provided in the Discussion section.
3. The authors mainly described the knowledges obtained from previous studies in the Discussion section. I recommend that the authors discuss the usefulness of DI reduction in chemotherapy for CRC, including the knowledges obtained from the results in the present study.
4. Please insert the results of p-values and ORs in the Figure 4.
Author Response
Dear Reviewer!
I'm really thankful about Your work, Your comments and corrections. This is the first time I try to publish a paper on international field. Your suggestions help me to improve my paper, which has not only professional, but personal importance to me.
Here are my answers:
- (P8L267) “Figure 6” should be changed to “Figure 5”.
The mistake is corrected in line 267.
- Limitations in this study should be provided in the Discussion section.
Which limitations do You think should be provided? You mean that it’s a single institue and retrospective data collection?
- The authors mainly described the knowledges obtained from previous studies in the Discussion section. I recommend that the authors discuss the usefulness of DI reduction in chemotherapy for CRC, including the knowledges obtained from the results in the present study.
With my co-author we had a discussion about this. The question was whether we should only present the data without suggestions or we should try to insert these knowledge into the everyday practice. Our temporary results suggests that maybe this classic schedule of de Gramont scheme is not the optimal use for the more toxic combined regimes. We are sure about that the cycle-time extension up to 21 days and the well scheduled drug holidays are improving PFS and surely not reducing OS. But upfront cycle-time extension is out of protocol, a deviation. This is why we are planning a clinical trial comparing mFOLFIRI based treatments with 14 vs. 21 days cycle time. The result of this trial – if it proves our presumptions – are enough to change the protocol to make the longer cycle-time officially available in everyday practice. Before that we think that the evidence that lower DI doesn’t reduce OS is enough to make our colleges be more flexible in conducting long term chemotherapy of mCRC patients.
- Please insert the results of p-values and ORs in the Figure 4.
The missing p-values and ORs inserted in Figure !.
If You have any more comments or suggestions, please inform me.
Thank You very much.
Yours faithfully,
Balázs Pécsi

Reviewer 2 Report
Review in Real-life effectivity of dose intensity reduction of first-line 2 mFOLFIRI-based treatment of metastatic colorectal cancers
Authors: Balázs Pécsi and LászlóMangel
General comment:
In this study the authors perform a real-life study to understand if palliative care of patients with metastatic colorectal cancer, using lower dose intensities, improves patient PFS and OS. The authors show that lower doses of FOLFIRI-based treatments are equally effective in CRC management but present lower toxicities to the patient, enabling prolonged treatments which lead to higher PFS and OS of mCRC patients. It was important to see that dose reduction did not reduce the effectiveness of the treatment.
Major comments
1. The authors state that It is important to see that dose reduction did not reduce the effectiveness of the treatment.However, it is not clear in the Figure 4 - Kaplan-Meier curves – if there is a clear benefit in mPFS / mOS with the different dose alterations. Do patients survive longer because treatments can be prolonged due to lower dosage?
2. In lines 248-250 the authors state that In the case of drug holidays, not only primary mPFS, but also mOS increased significantly. Among the analyzed DI reducing effects, drug holiday has the largest effect on mOS. This data is not clear on the graphs. Can the authors clarify?
3. There are many groups of patients with different treatment schemes. The description of all these groups is confusing and difficult to interpret. Maybe there should be a scheme to help define the groups of patients and which groups are being compared.
4. Is metastasectomy a reason to stop chemotherapy (line 133)? Please explain.
Minor comments:
1. Some concepts need to be better defined for non-clinicians. For example, what does “drug holiday” mean?
2. What is the difference between “dose reduction” and “DI reducing effect”?
3. The graphs in figures 2 and 3 are difficult to interpret. Additional legends should be provided to help on the data interpretation.
4. As the authors use many abbreviations, a list of abbreviations should be provided.
5. In line 125, the number of cycles should be in median form, with minimum and maximum values.
6. In table 1, what are the numbers in parenthesis after progression free survival and overall survival median time? It should be an interval.
7. What is the difference between intolerable toxicity and patients’ rejection of further chemotherapy (line 131)? It should be explained.
8. What were the toxicities related to stopp the treatment? Hematological? Other?
Author Response
Dear Reviewer!
I'm really thankful about Your work, about the many contructive suggestions You made. This is the first time when I try publish a paper on international field, I have no practise on this level, so any help, that improves my "product" has a big value to me.
Here are my answers:
Major comments
- The authors state that It is important to see that dose reduction did not reduce the effectiveness of the treatment.However, it is not clear in the Figure 4 - Kaplan-Meier curves – if there is a clear benefit in mPFS / mOS with the different dose alterations. Do patients survive longer because treatments can be prolonged due to lower dosage?
All data suggests that any kind of dose modification do not reduce mPFS or mOS. In mPFS all these changes cause significant improvement, in mOS the rate of improvement was not significant, but the trend showed better result, longer mOS. Upon these data we may not say the dose modification means longer survival, but we may say definitely that the dose reduction do not reduce survival. To achieve at least the same OS with less toxicity surely reduce the burden of patients caused by the treatment.
- In lines 248-250 the authors state that In the case of drug holidays, not only primary mPFS, but also mOS increased significantly. Among the analyzed DI reducing effects, drug holiday has the largest effect on mOS. This data is not clear on the graphs. Can the authors clarify?
Compared to the unchanged control group the result of cycle-time extension was median OS 791 days (p= .0526; 2ys OS OR 0.49 [95%CI 0.26-0.92]), the result of dose reduction was medias OS 773 days (p= .5562; 2ys OS OR 0.69 [95%CI 0.36-1.31]) and the result of drug holiday was median 975 days (p< .0001; 2ys OS OR 0.24 [95%CI 0.13-0.41]. The absolute number of mOS and the p-value was the highest in drug holiday group. These trend suggests that the drug holiday has the largest effect on mOS.
- There are many groups of patients with different treatment schemes. The description of all these groups is confusing and difficult to interpret. Maybe there should be a scheme to help define the groups of patients and which groups are being compared.
This paper contains a lot of data, a lot of comparison. The effect of cycle-time extension on the 3 different therapy scheme are compared on Figure 2. In the further comparisons the data of 3 different treatment groups were merged into 1 group where the different DI reducing effects were analyzed and compared. To reach the maximal volume of the article required strong reductions, maybe this compression led to the difficulties in understanding.
- Is metastasectomy a reason to stop chemotherapy (line 133)? Please explain.
Throughout the 1st line chemotherapy there is sometimes a chance to perform metastasectomy for better OS result. Metastasectomy means minimum 6 weeks off systemic therapy. In our protocol, after metastasectomy, achieving a “non target” situation we continue the treatment with “postmetastasectomic” FOLFOX scheme.
Minor comments:
- Some concepts need to be better defined for non-clinicians. For example, what does “drug holiday” mean?
The short explanation of the term „drug holiday” is complemented in the text at the first mentioning (line 107). Please, point if You find any other concepts that require better definition.
- What is the difference between “dose reduction” and “DI reducing effect”?
DI reducing effect is an overall term for the 3 DI reducing effects, like dose reduction, cycle time extension and drug holiday.
- The graphs in figures 2 and 3 are difficult to interpret. Additional legends should be provided to help on the data interpretation.
The lower part of Figure 2, was missing, I changed it to the original complete. The legends for all axis, the legends of the different curves are available. Please clarify, which legend is missing, what kind of information on the Figure would help to understand.
- As the authors use many abbreviations, a list of abbreviations should be provided.
The article draft is complemented by the suggested list of abbreviation at the end.
- In line 125, the number of cycles should be in median form, with minimum and maximum values.
Though the median number of cycles with minimum and maximum values of each treatment type groups are visable separately in Table 1. Line 3, the summarized data are complemented in line 125.
- In table 1, what are the numbers in parenthesis after progression free survival and overall survival median time? It should be an interval.
These numbers are the minimum and maximum days of mPFS, mPFS in each selected groups.
- What is the difference between intolerable toxicity and patients’ rejection of further chemotherapy (line 131)? It should be explained.
We consider that the treatment modification or cessation due to professional need is different to those in which professionally tolerable side-effects cause the patient’s rejection of therapy, though the treatment would be continuable.
- What were the toxicities related to stopp the treatment? Hematological? Other?
The major toxicities were haematological and intolerable side-effects, like long lasting vomitus despite anti-vomitus treatment or also therapy-resistant diarrhea. The short explanation is complemented in line 133.
Thank You very much for Your kind cooperation.
Yours faithfully,
Balázs Pécsi

Round 2
Reviewer 2 Report
Dear Author,
Thank you very much for clarifying. Would it be possible to integrate these explanations in the text of the paper, to facilitate the interpretation of the paper?
Many thanks in advance
Author Response
Dear Reviewer!
Thank You again for Your help and advices. According to Your suggestion in Round 2, I checked the original text and complemented IT by the missing explanations in Line 136-137 and 277-279. The others are already placed or replaced there.
Yours faithfully,
Balázs Pécs
